# Correlated anomalous phase diffusion of coupled phononic modes in a sideband-driven resonator

F. Sun[1], X. Dong[1], J. Zou[1], M.I. Dykman[2] & H.B. Chan[1]

The dynamical backaction from a periodically driven optical cavity can reduce the damping of a mechanical resonator, leading to parametric instability accompanied by self-sustained oscillations. Here we study experimentally and theoretically new aspects of the backaction and the discrete time-translation symmetry of a driven system using a micromechanical resonator with two nonlinearly coupled vibrational modes with strongly differing frequencies and decay rates. We find self-sustained oscillations in both the low- and high-frequency modes. Their frequencies and amplitudes are determined by the nonlinearity, which also leads to bistability and hysteresis. The phase fluctuations of the two modes show near-perfect anti-correlation, a consequence of the discrete time-translation symmetry. Concurrently, the phase of each mode undergoes anomalous diffusion. The phase variance follows a power law time dependence, with an exponent determined by the $1/f$-type resonator frequency noise. Our findings enable compensating for the fluctuations using a feedback scheme to achieve stable frequency downconversion.

[1] Department of Physics, The Hong Kong University of Science and Technology, Clear Water Bay, Kowloon, Hong Kong, China. [2] Department of Physics and Astronomy, Michigan State University, East Lansing, Michigan 48824, USA. Correspondence and requests for materials should be addressed to H.B.C. (email: hochan@ust.hk).

Advances in micro- and nano-mechanical resonators have made them indispensible tools for ultrasensitive detection of displacement, force, mass and charge[1–6]. By nonlinearly coupling them to optical or microwave cavities[7,8], the mechanical motion can be controlled through the interaction with electromagnetic field. The coupling creates sidebands around the cavity resonance frequency at $\omega_c \pm \omega_m$, where $\omega_c$ and $\omega_m$ are the resonance frequencies of the cavity and the mechanical resonator respectively (typically, $\omega_c \gg \omega_m$). When the cavity is driven at the red sideband $\omega_c - \omega_m$, the dynamical backaction leads to an increase in the damping and decrease in the effective temperature of the mechanical resonator[9–14]. Using sideband cooling, a number of groups have successfully cooled a mechanical resonator towards its quantum ground state[15–17]. Conversely, the backaction of the mechanical resonator leads to optomechanically induced transparency of the cavity[18,19]. For a weak driving at the blue sideband $\omega_c + \omega_m$, the mechanical resonator undergoes a decrease in damping and an increase in effective temperature, and the cavity exhibits optomechanically induced absorption[20]. When the driving amplitude increases beyond a threshold, the damping of the mechanical resonator becomes negative and a parametric instability develops[9,21]. In the steady state, self-sustained oscillations of the resonator take place[22–27], the amplitude of which is limited by nonlinear effects. For a different excitation mechanism, it has been predicted that nonlinearity can give rise to dynamical multistability[21], where several stable oscillation states coexist. Previous experiments on self-sustained oscillations mainly focused on the mechanical mode[22–28]. Alternatively, the transmission of cavities with linewidth comparable to the detuning of the pump laser was observed to be modulated by the vibrations of the mechanical mode[23]. Self-sustained oscillations of the cavity in the deep resolved sideband limit have remained largely unexplored.

There has also been much recent interest in the squeezing of thermal[29,30] or quantum noise[31,32] under non-degenerate parametric amplification. In the quantum case, correlations were found in the pulsed regime, whereas in the classical case correlations in the oscillation quadratures of the two modes were observed as the quadratures fluctuate about the mean value of zero[29,30]. Self-sustained oscillations, unlike the thermal or quantum fluctuations, are steady-state oscillations with non-zero amplitude and a well-defined phase. In the phase space of the oscillation quadratures, the equilibrium position of the system is displaced away from the origin. For two parametrically excited modes, the existence of correlations between the self-sustained oscillations has not been explored theoretically and experimentally.

In this work we consider a phonon[33] rather than a photon cavity, which is a part of a nonlinear electromechanical resonator. The cavity mode is nonlinearly coupled to a mechanical mode of the resonator at a lower frequency and with a smaller decay rate. The frequencies of the two modes differ by a factor $\approx 30$, and their decay rates differ by a factor of $\approx 61$. We study the self-sustained oscillations of the coupled modes under blue-detuned driving. The onset of the parametric instability is a Hopf bifurcation[34]. We observe that the number of stable states of the coupled modes changes with the driving frequency $\omega_F$. The change is accompanied by hysteresis. A crucial element of our experiment is that it is done in the sideband-resolved regime. We find that self-sustained oscillations are induced not only in the low-frequency mode as measured previously[22–27] but also in the high-frequency cavity mode. Our set-up allows simultaneous measurement of the amplitude and phase of both modes, which provides a means to reveal the discrete time-translation symmetry of the system. As expected, the phases of the self-sustained

oscillations fluctuate[35–37]. Remarkably, we find that the phase fluctuations of the two modes are strongly anti-correlated, so that the measured sum of the two phases remains constant within our detection limit. When external additive noise, which mimics thermal noise, is added to the system, the variance of the phase of each mode increases linearly with time in accordance with the standard phase diffusion picture. Without the external noise, the intrinsic mode eigenfrequency fluctuations (the frequency noise) are the dominating source of phase fluctuations. The variance of the phase increases with time superlinearly following a power law. We show that the corresponding exponent is determined by the exponent of the $1/f$-type frequency noise. The presence of frequency noise has recently become one of the crucial factors that affect the performance of mechanical resonators. A number of schemes were developed to isolate the contribution of frequency noise from thermal or detector noise[38–41]. Our results provide a novel way to identify and study frequency fluctuations via the phase diffusion of self-sustained oscillations. Furthermore, our findings of the strong phase anti-correlation of the two modes allow us to implement a feedback scheme that holds promise in significantly improving the phase stability of resonators in frequency standards and resonant sensing.

## Results

**The two-mode electromechanical system.** The resonator in our experiment is made of polycrystalline silicon. It consists of a plate (100 μm by 100 μm by 3.5 μm) supported on its opposite sides by two beams (1.3 μm wide and 2 μm thick, metalized with 30 nm of gold) of length 100 and 90 μm, respectively. Two vibrational modes are identified in our experiment. We call the first mode the plate mode. This mode has resonance frequency $\omega_1 = 198{,}780.9$ rad s$^{-1}$ and damping constant $\Gamma_1 = 1.87$ rad s$^{-1}$. It involves translational motion of the plate normal to the substrate, with both beams bending in this direction (lower inset in Fig. 1c). For the second mode, which we call the beam mode, only one of the beams (with length of 100 μm) vibrates, with displacement parallel to the substrate (lower inset in Fig. 1d). Since the plate remains stationary, the resonance frequency ($\omega_2 = 6{,}331{,}449$ rad s$^{-1}$) is significantly higher than that of the first mode. The damping constant is also higher, with $\Gamma_2 = 115$ rad s$^{-1}$. As shown in experiments on doubly clamped beams[33], the second mode can act as a phonon cavity mode, playing a similar role to photon cavity modes in optomechanical systems. In these previous experiments[33], however, the driving at the blue-detuned sideband was not sufficiently strong to induce self-sustained oscillations.

The parametric coupling between the plate mode and the phonon cavity (the beam mode) originates from the tension generated in the beam as it deforms parallel to the substrate, which in turn modifies the spring constant for the motion of the plate mode out of the substrate. Figure 1c shows that the resonance frequency of the plate mode increases linearly with the square of the amplitude of the beam mode[42,43]. Figure 1d plots the corresponding behaviour for the beam mode. In addition, as the plate vibrates, sidebands are created in the spectrum of the response of the phonon cavity around its frequency $\omega_2$. With $\omega_1/\Gamma_2 = 1{,}738$, parametric coupling between the plate mode and the beam mode leads to dynamical backaction effects in the deep resolved-sideband limit. To parametrically pump the system near the sidebands, a.c. current is applied to the beam in a magnetic field to generate a periodic Lorentz force at frequencies near $\omega_2 \pm \omega_1$. This force leads to a displacement of the beam, which results in the direct parametric modulation of the mode coupling and also in an indirect modulation through the nonlinear coupling between the modes (Supplementary Note 1).

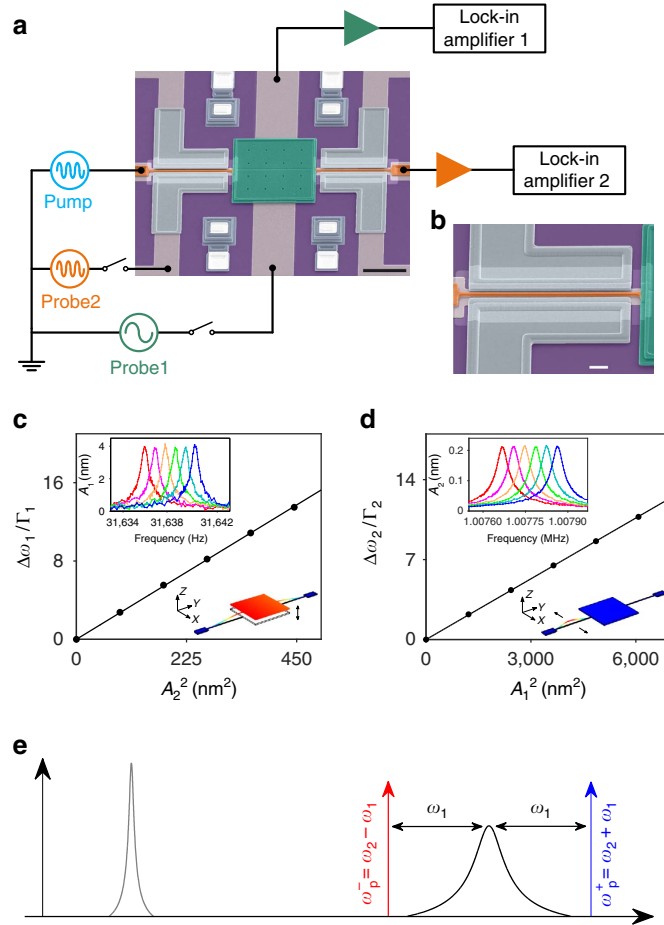

**Figure 1 | Dispersive coupling of the plate and beam modes of the electromechanical resonator.** (**a**) Scanning electron micrograph of the electromechanical resonator (colourized) and a schematic of the measurement circuitry. The black scale bar at the lower right corner measures 50 µm. (**b**) Close-up scanning electron micrograph of the beam. The white scale bar measures 10 µm. (**c**) The scaled shift of the resonance peak of the plate mode $\Delta\omega_1/\Gamma_1$ versus the vibration amplitude square of the beam mode $A_2$. The error bar is smaller than the dot size. The line is a linear fit. Upper inset: spectra of the plate mode measured with a small probe voltage (Probe1). $A_2^2$ is increased from zero in steps of 89 nm$^2$. Lower inset: vibration profile of the plate mode. (**d**) A similar plot for the beam mode. Upper inset: the squared plate mode amplitude $A_1^2$ is increased in steps of 1,215 nm$^2$. (**e**) A sketch of the spectra of the plate mode and the beam mode centred at the mode frequencies $\omega_1$ and $\omega_2$, respectively. The low- and high-frequency sidebands are marked as red and blue arrows, respectively.

The simplest equation of motion that captures the resonant behaviour is

$$\ddot{q}_1 + \omega_1^2 q_1 + 2\Gamma_1 \dot{q}_1 + (\gamma/m_1)q_1 q_2^2 + (\gamma_1/m_1)q_1^3 = (F/m_1)q_2 \cos \omega_F t$$
$$\ddot{q}_2 + \omega_2^2 q_2 + 2\Gamma_2 \dot{q}_2 + (\gamma/m_2)q_1^2 q_2 + (\gamma_2/m_2)q_2^3 = (F/m_2)q_1 \cos \omega_F t$$

$$(1)$$

where $q_1$ is the displacement of the plate, $q_2$ is the displacement of the midpoint of the beam, $m_{1,2}$ are the effective masses of the two modes, $\gamma$ denotes the dispersive coupling coefficient (the coupling energy is $\frac{1}{2}\gamma q_1^2 q_2^2$), $\gamma_{1,2}$ are the coefficients of the Duffing nonlinearity of the two modes and $F$ determines the amplitude of parametric pumping at frequency $\omega_F$ close to $\omega_2 \pm \omega_1$. The analysis applies also if, instead of parametric

driving, one considers driving the beam additively (second term in Supplementary equation (3)), which is closer to the conventional optomechanical set-up.

The coordinates $q_1$ and $q_2$ of our high-Q modes oscillate at frequencies $\omega_1$ and $\omega_2$, respectively, with slowly varying amplitudes and phases. Resonant dynamics of the coupled modes are conveniently described in the rotating frame. For the driving frequency close to the upper sideband frequency, $|\omega_F - \omega_1 - \omega_2| \ll \omega_{1,2}$, we change from $q_{1,2}(t), \dot{q}_{1,2}(t)$ to complex amplitudes, $q_1(t) = u_1(t)\exp(i\omega_1 t) + \text{c.c.}$, $q_2(t) = u_2(t)\exp[i(\omega_F - \omega_1)t] + \text{c.c.}$ In the rotating wave approximation, equations for $u_{1,2}(t)$ read

$$\dot{u}_1 + \Gamma_1 u_1 - i(\gamma_{12}/\omega_1)u_1|u_2|^2 - i(3\gamma_{11}/2\omega_1)u_1|u_1|^2 = (F_1/4i\omega_1)u_2^*,$$
$$\dot{u}_2 + (\Gamma_2 + i\Delta)u_2 - i(\gamma_{21}/\omega_2)u_2|u_1|^2 - i(3\gamma_{22}/2\omega_2)u_2|u_2|^2 = (F_2/4i\omega_2)u_1^*,$$

$$(2)$$

where $\Delta = \omega_F - \omega_1 - \omega_2$. Parameters $F_{1,2}$, $\gamma_{12}$, $\gamma_{21}$, $\gamma_{11}$ and $\gamma_{22}$ are determined by $F$, $\gamma$, $\gamma_1$ and $\gamma_2$, respectively, and also include contributions from the renormalization due to the coupling between the two modes (Supplementary Note 1). $\gamma_{12}$ ($4.6 \times 10^{22}$ rad$^2$ s$^{-2}$ m$^{-2}$) and $\gamma_{21}$ ($5.0 \times 10^{24}$ rad$^2$ s$^{-2}$ m$^{-2}$) are obtained by fitting the curves in Fig. 1c,d, respectively, whereas $\gamma_{11}$ ($7.0 \times 10^{21}$ rad$^2$ s$^{-2}$ m$^{-2}$) and $\gamma_{22}$ ($5.2 \times 10^{25}$ rad$^2$ s$^{-2}$ m$^{-2}$) are obtained by fitting the curves of nonlinear response to resonant driving.

In the case of driving close to the lower sideband frequency, $|\omega_F - \omega_2 + \omega_1| \ll \omega_{1,2}$, one sets $q_2(t) = u_2(t)\exp[i(\omega_F + \omega_1)t] + \text{c.c.}$ Equations for $u_{1,2}$ have the same form as equation (2), except that in the right-hand sides one should replace $u_2^* \rightarrow u_2, u_1^* \rightarrow u_1$ and also $\Delta = \omega_F + \omega_1 - \omega_2$.

**Dynamical backaction effects.** Figure 2a shows the resonance line-shape of the beam mode probed by applying a small periodic voltage on a nearby electrode (probe2 in Fig. 1a) at frequency $\omega_2 + \Delta\omega_2$. For red-detuned pumping at $\omega_F = \omega_2 - \omega_1$, a narrow dip appears on the spectrum. This dip is a manifestation of a photon-assisted Fano resonance where, in quantum terms, the phonon of the plate mode accompanied by a photon goes through the broad absorption band of the high-frequency mode. For mechanical resonators coupled to optical, microwave or phonon cavities, this feature is known as opto/electromechanically induced transparency[18,19,33]. Backaction on the plate mode leads to an increase in its damping and decrease in effective temperature. As shown in Fig. 2b, the damping of the plate mode increases with the pumping power at the red-detuned sideband. From equation (2), it follows that $\Gamma_1 \rightarrow \Gamma_1 + F_1 F_2/16\omega_1\omega_2\Gamma_2$ for $\omega_F = \omega_2 - \omega_1$ and $\Gamma_2 \gg \Gamma_1$. When the pumping frequency is changed to the upper sideband, the dip in the spectrum of the beam mode turns into a peak in Fig. 2a. Such enhanced absorption of the cavity mode is known as opto/electromechanically induced absorption in cavity optomechanics[20]. For blue-detuned pumping, the damping of the plate mode decreases with pumping power. From equation (2), $\Gamma_1 \rightarrow \Gamma_1 - F_1 F_2/16\omega_1\omega_2\Gamma_2$. In Fig. 2b, the two fitted straight lines have slopes that are nearly equal in magnitude but with opposite signs. There are small deviations from the above expressions due to additional heating effects as the pump power increases.

**Self-sustained oscillations and dynamical bistability.** When the blue-detuned drive amplitude is increased beyond a threshold value, the damping of the plate mode becomes negative. Fluctuations are amplified into self-sustained vibrations. Figure 2c,d shows the onset of such vibrations. They appear as

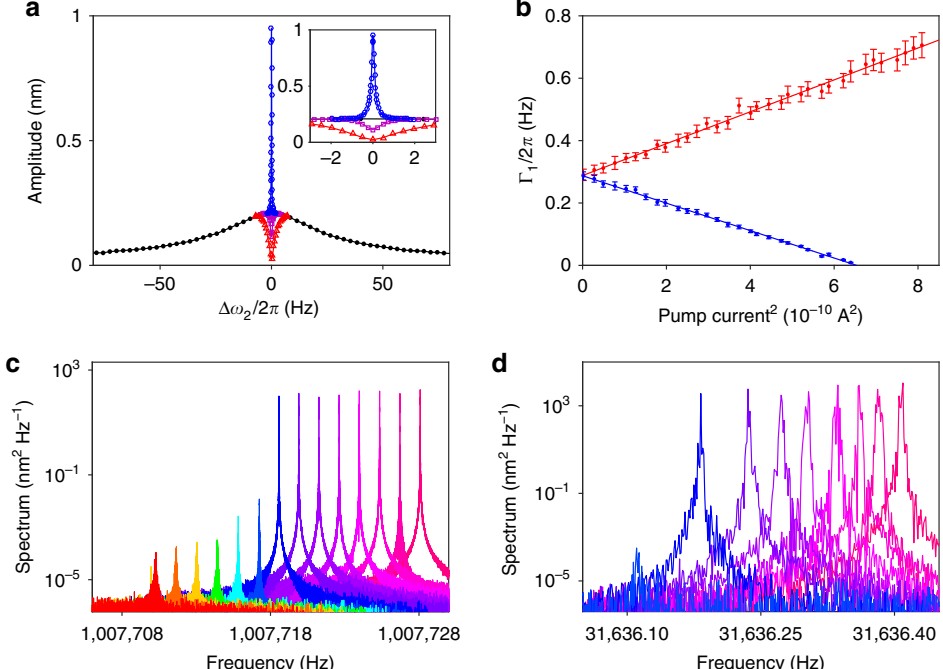

**Figure 2 | Dynamical backaction effects on the plate mode and the beam mode.** (**a**) Linear response of the beam mode probed by applying a small periodic voltage to a nearby electrode (probe2 in Fig. 1a) at frequency $\omega_2 + \Delta\omega_2$. The black curve corresponds to no pumping. For red-detuned pumping, a dip appears and the system exhibits induced transparency. For blue-detuned pumping, the dip turns into a peak (blue). Inset: expanded view of the extra peaks and the dip. The pumping currents are 21 and 70 μA for the purple and red data points, respectively, and 21 μA for the blue data points. (**b**) The dependence of the damping constant of the plate mode on the pumping power for red- and blue-detuned pumping. The lines are linear fits. Error bars represent 95% confidence intervals, obtained from the fit of the measured vibration amplitude of the plate as a function of the frequency of a small periodic force. (**c**) Blue-detuned pumping: power spectra of self-sustained vibrations of the beam mode for square of pumping current from 546 to 772 μA$^2$ in 17.4 μA$^2$ steps. The peak shifts with pumping current due to Duffing nonlinearity and dispersive coupling. (**d**) Power spectra of the plate mode. Spectra can only be resolved from the detection noise for pumping current $>633$ μA$^2$. The peak heights are only approximate because of frequency resolution limitations.

sharp increase in the height of the peaks in the power spectra of the plate and beam, respectively. While self-sustained vibrations of the low-frequency mechanical mode induced by sideband driving are well known[22–27], to our knowledge self-sustained oscillations of the cavity modes in the resolved sideband limit have not yet been studied in previous experiments. Figure 3a,b plots the square of the amplitude of self-sustained vibrations, for the plate and beam, respectively, as a function of the pump frequency detuning $\Delta = \omega_F - \omega_2 - \omega_1$. For $\Delta < -\omega_c$ only the zero-amplitude state is stable and no self-sustained vibrations take place. Here from equation (2)

$$\omega_c \approx (\Gamma_1 + \Gamma_2)(-1 + F_1 F_2 / 16\omega_1\omega_2\Gamma_1\Gamma_2)^{1/2}. \quad (3)$$

At $\Delta = -\omega_c$, there occurs a supercritical Hopf bifurcation with the onset of self-sustained vibrations. At $\Delta = \omega_c$, a subcritical Hopf bifurcation occurs. As $\Delta$ is swept down from $\Delta > \omega_c$, the vibration amplitudes jump sharply from zero to finite values at $\omega_c$. The frequency $\omega_c$ therefore serves as an important parameter that determines the range of detuning for bistable behaviour. Interestingly, it is independent of the nonlinearity parameters. Using equation (3), the proportionality constant between $(F_1 F_2)^{1/2}$ and the pumping current is found to be $2.6 \times 10^{12}$ rad s$^{-2}$ A$^{-1}$.

In the region $\Delta > \omega_c$, the system displays bistability. It has a stable state with no vibrations excited and a state with self-sustained vibrations. On the branch that corresponds to self-sustained vibrations, the vibration amplitude continues to increase with $\omega_F$ beyond $\omega_c$. Even though $\omega_F$ is comparatively far from $\omega_2 + \omega_1$, at such vibration amplitudes the dispersive coupling of the modes and the Duffing nonlinearities lead to an

increase in the sum of the amplitude-dependent mode frequencies, so that the resonant conditions can be satisfied. The frequencies of the self-sustained vibrations of the plate and beam modes are $\omega_1 + \delta\omega$ and $\omega_2 + \Delta - \delta\omega$, where $\delta\omega$ is given by Supplementary equation (7). They change with varying $\Delta$, but the vibrations remain nearly sinusoidal for not too large $\Delta$ and the sum of the vibration frequencies remains equal to $\omega_F$. In Fig. 3a,b, the solid lines are the non-trivial solutions of $|u_{1,2}|^2$ in equation (2) obtained by setting $u_1 \propto \exp(i\delta\omega t)$ and $u_2 \propto \exp(-i\delta\omega t)$. The slopes of the lines involve no adjustable parameters.

**Anti-correlated phase diffusion of the two modes.** Along with the amplitudes, an important characteristic of self-sustained vibrations are their phases[36]. In contrast to thermo-mechanical vibrations, self-sustained vibrations have well-defined phases in the sense that the phases vary on the timescale that largely exceeds not just the vibration period, but the relaxation times of the vibrational modes. These variations are random, representing phase fluctuations. The phase fluctuations determine the linewidths in Fig. 2c,d. There are two major sources of phase fluctuations. The first is thermal noise that is associated with the damping through the fluctuation–dissipation relation. The other is the direct fluctuations of the mode frequency, to be further discussed.

To analyse these fluctuations we write $u_n = |u_n| \exp[i(-1)^{n+1} \delta\omega t + i\phi_n(t)]$ ($n = 1, 2$). The phase changes of the modes are $\Delta\phi_{1,2}(t) = \phi_{1,2}(t) - \phi_{1,2}(0)$. From equation (2) we deduce that, in the absence of noise, the total phase $\phi_1 + \phi_2$ is uniquely determined by the driving field frequency and amplitude

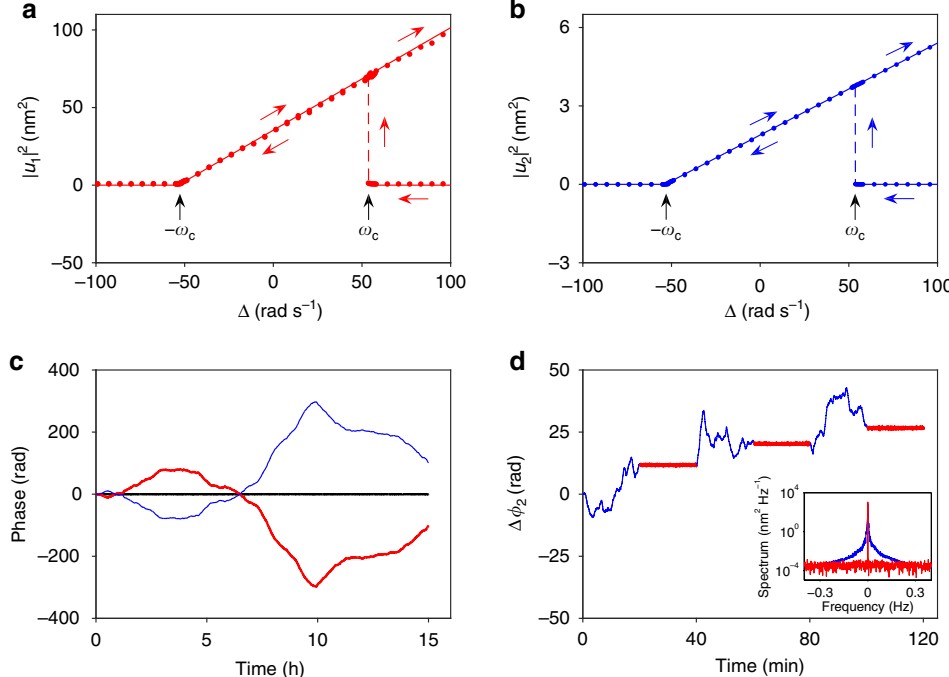

**Figure 3 | Hysteresis of self-sustained vibration amplitude and correlated phase fluctuations of the two modes.** (**a**) Square of the amplitude of self-sustained vibrations of the plate mode as a function of pump detuning $\Delta$. The error bar is smaller than the dot size. (**b**) The same plot for the beam mode. The solid lines are theoretical predictions using equation (2). The slopes involve no adjustable parameters. (**c**) Phase change for the plate $\Delta\phi_1(t)$ (red) and the beam mode $\Delta\phi_2(t)$ (blue). The black line shows $\Delta\phi_1(t) + \Delta\phi_2(t)$. The error bar is smaller than the line thickness. (**d**) Change in phase of the beam mode as a function of time. The feedback is turned on (off) for the red (blue) data. Inset: spectra of the self-sustained vibrations of the beam.

(Supplementary equation (9)). At the same time, the phase difference $\phi_1 - \phi_2$ can take on any value and the system has no rigidity with respect to $\phi_1 - \phi_2$. Therefore, in the presence of weak noise, fluctuations of $\phi_1 + \phi_2$ are small and do not accumulate in time, whereas fluctuations of $\phi_1 - \phi_2$ can accumulate. Figure 3c shows the measured $\Delta\phi_{1,2}(t)$ as functions of time, demonstrating that there is near perfect anti-correlation between the phase fluctuations of the two modes.

The strong anti-correlation between $\phi_1$ and $\phi_2$ opens the possibility of stabilizing the phase in one mode by feedback control, using the measured phase fluctuations in the other mode as an error signal. Figure 3d shows the phase of the beam mode as a function of time. With no feedback control (blue curve), the phase of the beam mode fluctuates in similar fashion as in Fig. 3c. For the red curve with feedback turned on, the phase of the plate mode is measured (by a lock-in amplifier with time constant of 0.5 s) and the phase of the pumping signal applied is adjusted to counter the change. Phase fluctuations in the beam mode are clearly reduced.

The ability to generate vibrations with low phase noise is of paramount importance in mechanical oscillators, enabling sensitive resonant detection and high stability precision clocks. Without sideband driving, feedback stabilization of phase has been shown for two modes in the linear regime for the case where their phase fluctuations originate from the common noise source[44]. In contrast, here the sources of phase fluctuations of the two modes can be independent from each other. The phase stabilization is based on the inherent phase anti-correlation and is achieved in a deeply nonlinear regime. The results enable tunable noise-free downconversion of the driving frequency $\omega_F$ to a frequency close to the eigenfrequency of the mechanical mode.

**Anomalous phase diffusion**. The phase noise of the self-sustained vibrations yields important information about the underlying mechanisms of fluctuations. As a test, we applied a broadband (200 Hz) electrical noise centred at $\omega_1$ to the bottom electrode of the plate. In Fig. 4a the red line shows that the mean square phase change of the beam mode $\langle\Delta\phi_2^2(t)\rangle$ increases linearly with time for such noise. The measured phase fluctuations are in good agreement with standard diffusion expected for a broadband noise. The slope of the linear dependence in Fig. 4a gives the diffusion constant $D$. Figure 4c shows that $D$ varies inversely with the square of the amplitude of self-sustained vibrations, in agreement with theoretical predictions[36,45]. As described earlier, the phase of the plate mode follows $\Delta\phi_1(t) \approx -\Delta\phi_2(t)$, in a manner similar to Fig. 3c. As the noise intensity or temperature increases, the phase fluctuations become stronger and $D$ is expected to increase.

When the external noise source is removed, the phase fluctuations change qualitatively. In Fig. 4a, the blue line shows that the mean square phase change no longer depends linearly on time. Instead, its time dependence is best fitted by a power law with exponent $\beta \sim 1.62 \pm 0.02$, as shown in the log-log plot in Fig. 4b. For other oscillation amplitudes, $\beta$ remains in the range from 1.6 to 1.8.

**Discussion**

A plausible source of the anomalous diffusion is frequency noise associated with the fluctuations of the eigenfrequencies of the modes. This phase diffusion is independent of the vibration amplitude. Intrinsic frequency noise was recently observed in silicon nitride, carbon nanotube and silicon mechanical resonators[38,39,41]. For resonators with high quality factors, frequency noise is of fundamental interest as it can become the limiting factor in the phase stability of vibrations. In our experiment, the anomalous phase diffusion arises mainly due to the frequency noise of the beam mode. Frequency noise of the plate mode is much smaller as its relatively large area allows for

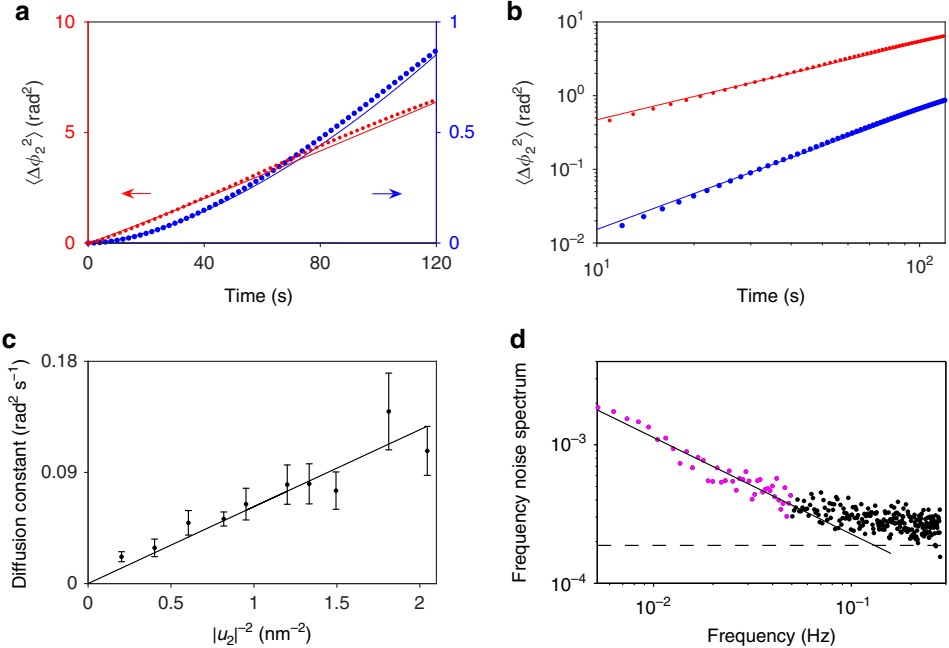

**Figure 4 | Anomalous phase diffusion and intrinsic frequency noise. (a)** When extra broadband noise is injected into the plate mode, the phase of the beam mode undergoes diffusion where the phase variance increases linearly with time (red). Without the extra noise, anomalous diffusion occurs as the time dependence becomes super-linear (blue). The error bar is smaller than the dot size. **(b)** Log-log plot of the same data. The slopes of linear fits (solid lines, also plotted in **a**) yield power law exponents of 1.05 and 1.62 for ordinary and anomalous diffusion, respectively. **(c)** The dependence of the phase diffusion constant $D$ in the presence of extra broadband noise on the inverse square oscillation amplitude. The line is a linear fit through the origin. Error bars represent $\pm 1$ s.e. **(d)** Power spectrum of the frequency noise of the beam mode. A fit to the purple region yields a dependence of the form $1/\omega^{0.7}$. The dashed line represents the minimum frequency noise that can be resolved, limited by noise in the detection circuits.

more efficient averaging of external disturbance such as forces due to trapped charges in dielectric layers. Figure 4d shows the spectrum of the frequency noise of the beam mode measured using the procedure described in ref. 39. We observe $1/f^{\alpha}$ behaviour of the spectrum at low frequencies with $\alpha = 0.7 \pm 0.07$. Such noise leads to phase fluctuations of the self-sustained vibrations that accumulate in time faster than for ordinary diffusion. The exponent $\beta$ in the time dependence of the anomalous phase diffusion is directly related to the frequency–noise exponent α:

$$\langle \Delta \phi_1^2(t) \rangle = \langle \Delta \phi_2^2(t) \rangle = C_f t^{\alpha+1} \qquad (4)$$

(constant $C_f$ is given in Supplementary Note 2). The exponent $\beta$ of the anomalous phase diffusion is in good agreement with equation (4), supporting the notion that it is the frequency noise that leads to anomalous phase diffusion of self-sustained vibrations. In our system, the frequency noise likely originates from the capacitive coupling of the resonator to the charge traps in the silicon nitride layer that electrically isolate the polysilicon structures from the silicon substrate. Further experiments to characterize the frequency noise are in progress.

Our findings show that, by driving a nonlinear resonator, one can simultaneously excite coupled modes with strongly different frequencies and decay rates. This excitation results from the interplay of nonlinearity and dynamical backaction. The amplitudes and frequencies of the vibrations depend on the strength and frequency of the driving and display hysteresis. Our central observation is the anomalous strongly anti-correlated diffusion of the vibrational phases of the modes, which is due to fluctuations of the mode eigenfrequencies. Besides being interesting on its own, this diffusion enables probing the frequency noise of oscillation modes even when this noise is weak. The anti-correlation of the phase fluctuations of the two

modes allows manipulation and stabilization of one phase by using the measured other phase deep into nonlinear regime.

The phase anticorrelation is a consequence of the discrete time-translation symmetry imposed by the periodic modulation. Because of this symmetry, the system reproduces itself on average every modulation period $2\pi/\omega_F$. The frequencies of the vibrations of each mode are generally incommensurate with $\omega_F$, and therefore the small-amplitude displacements $q_{1,2}(t)$ averaged over the period $2\pi/\omega_F$ are equal to zero and the phases of the individual modes are not fixed. However, the product $q_1(t)q_2(t)$ has a term oscillating at frequency $\omega_F$. The phase of this term is $\phi_1(t) + \phi_2(t)$. The time-translation symmetry requires that this phase remain constant on average, which eliminates accumulation of fluctuations of $\phi_1(t) + \phi_2(t)$ and thus diffusion of $\phi_1(t) + \phi_2(t)$. Our system is perhaps the simplest nontrivial system that displays self-sustained vibrations with different frequencies where the discrete time-translation symmetry can be revealed via phase measurements. It appears plausible that the discrete time-translation symmetry affects also the dynamics of three vibrational modes in a periodically driven electromechanical resonator where phonon lasing was recently demonstrated[46]. The detailed theory, however, will need to be developed.

While our experiment uses a micromechanical resonator coupled to a phonon cavity, the analysis can be extended to optical and microwave cavities in the resolved sideband limit. Understanding the underlying physics of frequency noise and minimizing phase noise can lead to further improvements of the performance of these cavities. Overall, the observed self-sustained vibrations offer new opportunities for signal transduction, tunable noise-free frequency downconversion, phase manipulation and investigation of fundamental properties of resonating cavities including those related to the discrete time-translation symmetry in driven systems.

## Methods

**Transduction scheme.** The device is placed in a magnetic field of 5 T perpendicular to the substrate, at a temperature of 4 K and pressure of $< 10^{-5}$ torr. For the beam mode, vibrations can be excited by applying an a.c. probe voltage at frequency close to $\omega_2$ to an electrode next to the beam (Probe2 in Fig. 1a). In-plane motion of the beam is detected by measuring the current induced by the electromotive force in the magnetic field. The other beam that supports the plate has resonance frequency far from $\omega_2$ and is not excited. For detecting motion of the plate, two fixed electrodes are fabricated underneath it. An a.c. probe voltage (Probe1 in Fig. 1a) at frequency close to $\omega_1$ applied to one of these electrodes (the lower one in Fig. 1a) can be used to capacitively generate a periodic electrostatic force. Vibrations of the top plate are detected by measuring the capacitance between it and the other electrode.

**Measurement of the mean square phase change.** The mean square phase change of the self-sustained oscillations in Fig. 4a is calculated from records of the phase over 24 and 6 h for the red and blue curves, respectively. More averaging was performed for the red curve due to the larger fluctuations. In both cases, segments $[\phi_2(t)]_k$ of 10 min each ($k = 1, 2, \dots$) are taken from the record of the phase of mode 2 (the beam). The segments are shifted in time by 1 s so that a 24 h record yields $\sim 85,800$ segments. Each segment is then fitted with a straight line, the slope of which gives the mean oscillation frequency $[\omega_2]_k$ for the 10 min interval. The phase fluctuations of each segment about $[\omega_2]_k$ is given by $[\Delta\phi_2(t)]_k = [\phi_2(t)]_k - [\phi_2(0)]_k - [\omega_2]_k t$, the square of which is averaged over $k$ to yield the mean square phase change as a function of time.

**Measurement of the frequency noise spectrum.** The frequency noise spectrum of the beam (Fig. 4d) is detected by applying periodic excitation near $\omega_2$ and extending the method of ref. 39. With the pump frequency adjusted to be far from both sidebands, an a.c. voltage is applied to a nearby gate to exert a periodic electrostatic force on the beam. In the absence of frequency noise, the a.c. force excites oscillations of the beam only at the driving frequency $\omega_d$. In the linear regime, this induced vibration is superimposed on the random thermal motion. For small and slow frequency fluctuations (with correlation time $t_r \gg 1/\Gamma_2$), the change in the amplitude and phase of the forced vibrations at $\omega_d$ is proportional to the shift in the eigenfrequency. In this limit, the spectrum of this change gives the frequency noise spectrum.

To reveal the frequency noise in our system, it is necessary to choose a driving amplitude that is sufficiently strong to overcome the noise in our detection circuits, which makes vibrations nonlinear, in contrast to the regime studied in ref. 39. In Fig. 4d, the dashed line represents the minimum frequency noise that can be resolved, limited by detection noise. This value is obtained by setting the driving frequency $\omega_d$ far from $\omega_2$ while keeping the driving amplitude fixed. Away from resonance, the amplitude of the forced vibrations decreased significantly and became buried by the detection noise. The spectrum that arises solely from the detection noise is measured to be flat for the range of frequencies in Fig. 4d.

**Data availability.** The data that support the findings of this study are available from the corresponding author on request.

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

## Acknowledgements

This work is supported by the Research Grants Council of Hong Kong SAR (Project No. 16315116). M.I.D. is supported by U.S. Army Research Office (W911NF-12-1-0235) and the National Science Foundation (DMR-1514591).

## Author contributions

H.B.C. and J.Z. conceived the experiments; M.I.D. developed the theory; F.S. and X.D. performed the experiments and analysed the data; F.S., H.B.C. and M.I.D. co-wrote the paper.

## Additional information

**Competing financial interests:** The authors declare no competing financial interests.

