## [Peer review file · Nature Communications]

Reviewers' Comments:

Reviewer #1 (Remarks to the Author)

Sun et al. have exploited the non-linear coupling between modes in a micromechanical system to explore self-sustained mechanical oscillations, focusing in particular on the phase dynamics. They reveal locking of the total phase, demonstrate stabilization of phase fluctuations using feedback and also uncover anomalous diffusion in the relative phase.

There have been a number of studies of self-sustained mechanical oscillations in optomechanical systems where there is a non-linear coupling between optical and mechanical modes; similar self-sustained oscillations have also been generated using nonlinear couplings between three mechanical modes [Mahboob et al., PRL 110, 127202]. However, the generation of self-sustained oscillations using just two coupled mechanical modes has not been demonstrated before. Furthermore, the detailed exploration of the phase dynamics in the present paper is extremely interesting and novel. This work is an important step forward in the area and is likely to inspire considerable further experimental and theoretical work. The methodology and analysis in the paper both appear to be entirely sound. For these reasons, I therefore feel that the paper is suitable for publication in Nature Communications.

However, before publication the authors should address the following points:

-They should discuss the work on 'phonon lasing' by Mahboob et al [PRL 110, 127202]. What are the advantages of their two mode scheme over the three mode approach? How do the results compare?

-The authors should also compare their work more closely with optomechanical studies which also looked at the dynamics of the cavity mode. For example, the cavity oscillations revealed in Ref 23 (see eg Fig 3), unlike those seen in the 'cavity' mechanical mode by the authors, do not undergo periodic oscillations with a fixed amplitude. What is the reason for this difference? Does it follow that the phase dynamics will also be quite distinctive from optomechanical systems?

Reviewer #2 (Remarks to the Author)

The authors employed a coupled nonlinear electromechanical resonator and excited self-oscillations in two resonant modes by pumping their blue side band. Each phase of these parametric oscillations is random but the phase of one mode always reveals an opposite sign to that of the other mode. This effect enables them to implement feedback control that suppresses the phase fluctuation of the beam mode, and to try exploring physical origin of temporal phase diffusion. As indicated, the observation of this anti-correlated phase phenomenon is their most important result in this study and it can be also seen from page 3 "Remarkably, the phases of the two modes are strongly anti-correlated, so that...within our detection limit" and page 13-14 "Our central observation is the anomalous strongly anti-correlated diffusion ...of the mode eigenfrequencies". They concluded that these findings are indispensable to improve performance of hybrid cavity systems and to study the fundamental physics, which can be seen from "Overall, the observed self-sustained vibrations offer new opportunities for signal transduction, frequency downconversion, phase manipulation and investigation of fundamental properties of resonating cavities".

However, I think this similar phase correlation has already been reported in several groups using an optomechanical system [1], an electromechanical system [2] and even in pure mechanical systems [3,4], in which pumping a blue side band of low- and high-frequency cavities gives rise to self-oscillations, resulting in the realization of two mode squeezed states, namely the phase of

these cavities are correlated each other. If the authors claim and justify the novelty and significance to their work, they should carefully explain the originality in the manuscript. On the other hand, I think their developed feedback technique that can reduce phase noise is interesting and it has originality with respect to other reports. I recommend the authors to more focus on this topic. Additionally, I also give several comments and questions concerning this manuscript as shown below. Unless they scientifically justify and do an appropriate modification to the above and below points, I think this manuscript is more suitable to go to applied and technological journals.

1. The authors often mentioned "Hopf bifurcation" in the manuscript. Is this terminology equal to self-sustained oscillation? It might confuse non-expert readers and so is desirable to describe it in more detail or to cite an appropriate journal.
2. In page 8, the authors have the section "Self-sustained oscillations and dynamical multistability". However, they here describe the observation of only bistable states in these modes. Usually I think that when we say "multi" we consider many vibration states far beyond 2. To avoid reader's misunderstanding, the section title should be modified.
3. Why does the anomalous phase diffusion come only from the high frequency beam mode? How is the plate mode? The detail explanation is desired.
4. q_1 and q_2 in the right hand side of equations (1) should be reversed.
5. I think the usages of anti-Stokes and Stokes are used in incorrect way and they should be opposite. The anti-Stokes and Stokes pumps should correspond to the blue and red side bands respectively.
6. In page 7, the authors describe that the red side pump creates a dip in the beam spectrum, enabling phonon to be transferred from the plate to the beam. However, in the measurement of Fig. 2a, phonon energy is put into the beam mode for probing. Therefore it should enable phonons to be transferred from the beam to the plate.
7. Why do the amplitudes of both self-oscillations saturate in spite of an increase in the pump current in Figs. 2c and 2d? What is the physical or technical reason?
8. The authors mentioned "The phase fluctuations determine the linewidths..." in page 10, but later, they also mentioned "..., the anomalous phase diffusion arises mainly due to the frequency noise ..." in page 13. It looks the relation of cause and effect is swapped in the manuscript and it will make the readers confused.
9. The meaning of the final sentence in page 11 "As described earlier, the phase of the plate..." is unclear. I recommend to put a further explanation here.
10. In the section of "Discussion", the authors speculate the anomalous phase diffusion is caused by the charge traps in the silicon nitride layer. But that is well known physics to describe $1/f$ noise. In this sense, although the noise is dominated by unusual $(1/f)^{0.7}$, it becomes a trivial phenomenon and no "anomalous". If the authors want to use anomalous in the phase diffusion process in this manuscript, it is desirable to more clarify the underlying mechanism of the $(1/f)^{0.7}$ noise.

[1] R. Riedinger et al., Nature 530, 313 (2016)

[2] T. A. Palomaki et al., Science 342, 710 (2013).

[3] I. Mahboob et al., PRL 113, 167203 (2014)

[4] Y. S. Patil et al., PRL 115, 017202 (2015).

Reviewer #3 (Comments to the Author)

Correlated anomalous phase diffusion of coupled photonic modes in a sideband drive resonator

Summary:

This paper presents a series of findings and observations regarding two coupled modes of a mechanical structure (*plate* and *beam* modes) as the coupled system is driven by an external harmonic force tuned to the Stokes and anti-Stokes sidebands ($\omega_p = \omega_1 \pm \omega_2$) of the high frequency mode (beam mode). The findings can be summarized as:

- 1- When $\omega_p = \omega_2 - \omega_1$ electromagnetically induced transparency appears in the spectrum of the high frequency mode and damping of the low frequency mode increases.
- 2- When $\omega_p = \omega_1 + \omega_2$ a peak appears in the spectrum of the high frequency mode and damping of the low frequency mode decreases.
- 3- When $\omega_p = \omega_1 + \omega_2$ and power is more than certain value (threshold) both modes oscillate.
- 4- The theoretical results for oscillation amplitudes versus pump detuning are in agreement with experimental results.
- 5- Observation of bistable behavior of oscillation amplitudes.
- 6- Observation of Anti-correlated phase diffusion of the two modes
- 7- Observation of linear and anomalous phase diffusion in the phase of the high frequency mode in the presence and absence of external noise injected to the low-frequency mode.

Among these 3, 6 and 7 seem to be new (not observed in phononic and phononic/photonic systems). The selected system is interesting and the quality of the results show the high accuracy of the measurement system. However these results don't seem to add up coherently into a strong claim that makes the paper stand out compared to previous work on a very similar system (in particular Ref. 29). Here are examples of recommendations that can make the paper stronger:

Abstract

-It is not clear from the abstract whether this is theoretical, experimental study or both. Using phrases like "Here we study", "We show", "these findings reveal" without specifying the nature of the study is confusing. By writing the abstract one should get a clear idea about what is presented in the paper. The main contribution of the work and its relevance are not clearly stated

Page-3) Last paragraph

- It is not clear what are the new findings compared to Ref. 29.
- Following sentences either express well-known facts (without a connection/relation to the objective of the paper) or are unclear:
 - ".. the phases of the self-sustained oscillation fluctuate."
 - "..our setup allows simultaneous measurement of the oscillation of both modes..". How? Why is this relevant if each is connected to a separate actuator? How is your setup different than others?

“Without the external noise, the intrinsic eigenfrequency fluctuations of high-frequency (cavity) mode are the dominating source of phase fluctuations”, phase fluctuations of the plate or the beam mode? (for the beam mode this is obvious).

The two-mode electromechanical system

-In Figure 1, how the beam and plate modes have been excited? The ac signal is applied on which port (pump, probe1 or probe2) and at what frequency? The general explanation presented in Methods-Transduction scheme does not provide sufficient information.

-“The parametric coupling between the plate mode and the phonon cavity are mediated by the tension in the beam as it deforms” What is the meaning of “mediation” here? Is this the same property explained in Ref. 29 (page 388 first sentence of the second paragraph: ...the motion of the first mode induces tension that can modify the frequency of the phonon cavity,...” Or a different mechanism is claimed?

-Here and throughout the two modes are referred to as high and low frequency modes, beam and plate modes and cavity mode. Please after the Figure-1 (where explain all terms) pick one set of words and stick to it.

-Clearly specify to which electrode (input port) you apply the ac current (instead of saying “applied to the beam” which implies exciting the beam mode.

-Clarify this sentence “.. this force leads to a displacement of the beam, changing the tension and hence ...”

-Describe the physical nature of what you call “dispersive coupling coefficient” in the coupled system under test.

-What is the meaning of “driving the beam additively”?

Results

-Page 8 near the end of first paragraph: “the peak shifts with pumping current due to nonlinear effects”. Which nonlinear effects? Is this referring to nonlinear coupling between the two modes? The observed effect seems to be a manifestation of the so called “optical spring effect” in optomechanical resonator/oscillators. Is that the case who is this observation is related to the same effect in those systems?

-What is physical relevance of ω_c (that only depends on damping and the amplitude of the pump) that makes it a critical parameter for bistable behavior?

-It seems that one of the new observation here (and according to abstract) is self-sustained oscillation of both modes when stokes pump power is beyond certain value. As such a clear discussion about this result and its relevance is needed. Why this was not observed before? What characteristic of the current system makes this possible.

-Why is it important to stabilize one mode using the feedback from measuring the phase fluctuations in another mode.

-As at room temperature thermal noise is the dominant noise mechanism a discussion about expected behavior at room temperature is missing.

End of Page 11) Why it has been assumed that even in the presence of external noise, the phases of the two modes are still anti-correlated? ‘

OVERALL CONCLUSION

The idea of using the coupled mechanical modes to explore the physical phenomena that were originally observed and explained in the optomechanically coupled systems (i.e. side-band cooling, parametric oscillation, bistable behavior and electromagnetically induced transparency) was proposed and studied in Ref. 29. However Ref. 29 was mainly focused on cooling while the results presented here are mainly related to oscillation and noise. The authors have carefully explored different aspects of oscillatory behavior of two coupled mechanical modes and reported some results that are interesting and relevant for future work in this area. In terms of the content, the paper is a mixture of well-known effects observed in a new system and few new observations (at least to my knowledge) especially in the behavior of the noise. As such the novelty factor and potential impact of the paper are not strong enough to automatically qualify it as Nature Communications.

The main weakness of the paper is that, at least in its current form, it appears as a characterization report with some physical insight. Several observations (some new and some just verification of previous work) have been presented in a scattered manner making the reader wonder about the main contribution of the authors. As such independent of its content the paper also requires major revision in writing style to become more accessible and understandable. I recommend the publication of the manuscript only if in a new revision the authors can clearly express the logic behind each experiment, relevance of their results and more importantly novelty of their approach/study compared to past work.

Response to Reviewer 1

We are pleased that Reviewer 1 finds our exploration of the phase dynamics “extremely interesting and novel”. We are grateful to her/him for the insightful comments and for recommending publication of our paper after minor revisions. We now address his/her comments.

Reviewer 1 indicates that, “before publication the authors should address the following points”.

1. They should discuss the work on ‘phonon lasing’ by Mahboob et al [PRL 110, 127202]. What are the advantages of their two mode scheme over the three mode approach? How do the results compare?

We have added the suggested reference, Ref. 46. We explain that the features of our dynamics are related to the discrete time-translation symmetry of the coupled vibrations in our system. To emphasize the difference of Ref. 46 from our work we indicate “Our system is perhaps the simplest nontrivial system that displays self-sustained vibrations with different frequencies where the discrete time translation symmetry can be revealed via phase measurements. It appears plausible that the discrete time translation symmetry affects also the dynamics of three vibrational modes in a periodically driven electromechanical resonator where phonon lasing was recently demonstrated⁴⁶. The detailed theory, however, will need to be developed.”

This is one example where our paper could inspire further investigations, as pointed out by Reviewer 1.

2. -The authors should also compare their work more closely with optomechanical studies which also looked at the dynamics of the cavity mode. For example, the cavity oscillations revealed in Ref 23 (see eg Fig 3), unlike those seen in the ‘cavity’ mechanical mode by the authors, do not undergo periodic oscillations with a fixed amplitude. What is the reason for this difference? Does it follow that the phase dynamics will also be quite distinctive from optomechanical systems?

To address the comment, we indicate in the text: “Alternatively, the transmission of cavities with linewidth comparable to the detuning of the pump laser was observed to be modulated by the vibrations of the mechanical mode²³. Self-sustained oscillations of the cavity in the deep resolved sideband limit have remained largely unexplored.” In the experiment [23], the cavity linewidth is comparable to the detuning of the incident light, and the phases of the modes could not be accessed. Moreover, much of the results refer to a regime where the mechanical vibrations are strongly nonlinear, with multiple overtones, all the way to the regime of dynamical chaos, so that a phase is not a good characteristic all together. We expect the phase dynamics observed in our system to also occur in optomechanical systems, provided they are in the resolved-sideband limit.

Response to Reviewer 2.

We are grateful to Reviewer 2 for the thorough review and for the advice to carefully explain the originality of the manuscript and put more emphasis on the new mechanism of phase stabilization, as well as for his/her comments and questions. We have followed the advice and made several changes, starting with the abstract and all the way to the conclusions. We now address the specific comments.

The main criticism of Reviewer 2 is that he/she thinks “similar phase correlation has already been reported in several groups using an optomechanical system [32], an electromechanical system [31] and even in pure mechanical systems [29, 30]”. However, the above experiments did not involve self-sustained oscillations and it is not possible to observe phase diffusion under those experimental conditions. We have added the following paragraph and the four references:

“There has also been much recent interest in the squeezing of thermal^{29,30} or quantum noise^{31,32} under non-degenerate parametric amplification. In the quantum case, correlations were found in the pulsed regime, whereas in the classical case correlations in the oscillation quadratures of the two modes were observed as the quadratures fluctuate about the mean value of zero^{29,30}. Self-sustained oscillations, unlike the thermal or quantum fluctuations, are steady-state oscillations with non-zero amplitude and a well-defined phase. In the phase space of the oscillation quadratures, the equilibrium position of the system is displaced away from the origin. For two parametrically excited modes, the existence of correlations between the self-sustained oscillations has not been explored theoretically and experimentally.”

We are pleased that Reviewer 2 believes that our “developed feedback technique that can reduce phase noise is interesting and it has originality with respect to other reports. I recommend the authors to more focus on this topic.” We stress that without the well-defined phases, the feedback technique described in our paper cannot be implemented. We have put more emphasis on the relation of the stabilization mechanism to the discrete time-translation symmetry and on the advantageous features of the proposed mechanism for frequency downconversion. In particular, we indicate that “The ability to generate vibrations with low phase noise is of paramount importance in mechanical oscillators, enabling sensitive resonant detection and high stability precision clocks. The results enable tunable noise-free downconversion of the driving frequency ω_F to a frequency close to the eigenfrequency of the mechanical mode.”

1. The authors often mentioned "Hopf bifurcation" in the manuscript. Is this terminology equal to self-sustained oscillation? It might confuse non-expert readers and so is desirable to describe it in more detail or to cite an appropriate journal.

As suggested by Reviewer 2, we have added a citation on Hopf bifurcation (Ref. [34]).

2. In page 8, the authors have the section "Self-sustained oscillations and dynamical multistability". However, they here describe the observation of only bistable states in these modes. Usually I think that when we say "multi" we consider many vibration states far beyond 2. To avoid reader's misunderstanding, the section title should be modified.

We are grateful to Reviewer 2 for spotting this inconsistency. The section title of "dynamical multistability" has been changed to "dynamical bistability"

3. Why does the anomalous phase diffusion come only from the high frequency beam mode? How is the plate mode? The detail explanation is desired.

We have added the following explanation:

"Frequency noise of the plate mode is much smaller as its relatively large area allows for more efficient averaging of external disturbance such as forces due to trapped charges in dielectric layers."

4. q_1 and q_2 in the right hand side of equations (1) should be reversed.

We have double checked that q_1 and q_2 on the right hand side of Eq. (1) are correct. Eq. (1) is consistent with the driving term in the Hamiltonian, $F q_1 q_2 \cos \omega t$ (Eq. S6 in supplementary information).

5. I think the usages of anti-Stokes and Stokes are used in incorrect way and they should be opposite. The anti-Stokes and Stokes pumps should correspond to the blue and red side bands respectively.

We have replaced all Stokes and anti-Stokes pump by driving at the lower and upper sidebands.

6. In page 7, the authors describe that the red side pump creates a dip in the beam spectrum, enabling phonon to be transferred from the plate to the beam. However, in the measurement of Fig. 2a, phonon energy is put into the beam mode for probing. Therefore it should enable phonons to be transferred from the beam to the plate.

To avoid confusion, we have modified the text to describe the dip in the response of the beam as a photon-assisted Fano resonance.

“This dip is a manifestation of a photon-assisted Fano resonance where, in quantum terms, the phonon of the plate mode accompanied by a photon goes through the broad absorption band of the high-frequency mode.”

7. Why do the amplitudes of both self-oscillations saturate in spite of an increase in the pump current in Figs. 2c and 2d? What is the physical or technical reason?

Reviewer 2 is correct in pointing out that the amplitude of self-sustained oscillations should not saturate. Since the spectra have a small width (much smaller than the inverse damping), capturing both the shape and the exact height of each peak requires very fine frequency resolution (and long measurement times). Figures 2c and 2d were each obtained over about 5 hours. While they serve the main purpose of demonstrating the onset of self-oscillations, the frequency resolution is not sufficient to display the maxima of the peaks. The maxima of the peaks were measured in a different run and plotted in Fig. 3. We have added the text “The peak heights are only approximate because of frequency resolution limitations.”

8. The authors mentioned "The phase fluctuations determine the linewidths..." in page 10, but later, they also mentioned "..., the anomalous phase diffusion arises mainly due to the frequency noise ..." in page 13. It looks the relation of cause and effect is swapped in the manuscript and it will make the readers confused.

We would like to clarify that there is no cause-effect relationship between these two sentences. Even when the eigenfrequency of the mode is fixed and stable, the phase of self-sustained vibrations fluctuates due to thermal effects. The phase diffusion leads to a non-zero linewidth for the spectrum of self-sustained oscillations. If, in addition, there are fluctuations in the eigenfrequency of the mode, the phase change increases more rapidly with time to display anomalous phase diffusion and further broadens the spectrum, as compared with ordinary phase diffusion when fluctuations in the eigenfrequency is absent. We have added the text “There are two major sources of phase fluctuations. The first is thermal noise that is associated with the damping through the fluctuation-dissipation relation. The other is the direct fluctuations of the mode frequency, to be further discussed.”

9. The meaning of the final sentence in page 11 "As described earlier, the phase of the plate..." is unclear. I recommend to put a further explanation here.

The phase fluctuations of the plate follow that of the beam with a negative sign. We have adopted the recommendation of Reviewer 2 and extended the sentence to include: “in a manner similar to Fig. 3c.”

10. In the section of "Discussion", the authors speculate the anomalous phase diffusion is caused by the charge traps in the silicon nitride layer. But that is well known physics to describe $1/f$ noise. In this sense, although the noise is dominated by unusual $(1/f)^{0.7}$, it becomes a trivial phenomenon and no "anomalous". If the authors want to use anomalous in the phase diffusion process in this manuscript, it is desirable to more clarify the underlying mechanism of the $(1/f)^{0.7}$ noise.

We use the term "anomalous diffusion" in the conventional way to indicate that the variance of the phase fluctuations is nonlinear in time. As shown by the blue line in Fig. 4a, we clearly observed anomalous diffusion in our system. The word "anomalous" does not refer to the exponent of the $1/f$ type noise. While we agree that the underlying mechanism of the $1/f$ frequency noise is an interesting topic, determining the precise origin of this noise is beyond the scope of this paper.

Response to Reviewer 3

We thank Reviewer 3 for the thoughtful suggestions. We are pleased that Reviewer 3 finds that we have new results and that "the selected system is interesting and the quality of the results show the high accuracy of the measurement system." Reviewer 3 expressed a concern that "these results don't seem to add up coherently". To address this concern we have modified the paper to emphasize that "our system is perhaps the simplest nontrivial system that displays self-sustained vibrations with different frequencies where the discrete time translation symmetry can be revealed via phase measurements"

We have incorporated all the recommendations of Reviewer 3 to make our paper stronger. We believe that the modified manuscript convey the novelty of our findings and the potential impacts.

Our itemized responses are listed below:

Abstract

-It is not clear from the abstract whether this is theoretical, experimental study or both. Using phrases like "Here we study", "We show", "these findings reveal" without specifying the nature of the study is confusing. By writing the abstract one should get a clear idea about what is presented in the paper. The main contribution of the work and its relevance are not clearly stated.

Following the suggestion of Reviewer 3, we indicated in the abstract that our work is an "experimental and theoretical study". We have significantly revised the abstract to state the main contribution of our work and its relevance.

Page-3) Last paragraph

-It is not clear what are the new findings compared to Ref. 29

Our paper focuses on phase fluctuations of self-sustained oscillations induced by strong blue-detuned sideband pumping. As Reviewer 3 mentioned later in the report, self-sustained oscillations were not observed at all in Ref. 29. We have added the sentence, "In these previous experiments, however, the driving at the blue-detuned sideband was not sufficiently strong to induce self-sustained oscillations." Here we summarize our findings that were not reported in Ref. 29, nor any other papers to our knowledge:

- i) Observation of self-sustained vibrations, which are induced both in the cavity mode and the mechanical mode.
- ii) Measurement of phase fluctuations as a means to reveal the discrete time-translation symmetry in a system of self-sustained vibration with incommensurate frequencies.
- iii) Observation of strong anticorrelation of the phase fluctuations and application of this effect for highly stabilized frequency downconversion.
- iv) Observation of the anomalous phase diffusion of self-sustained vibrations, which follows a superlinear power law with the exponent determined by the mode frequency fluctuations.

- Following sentences either express well-known facts (without a connection/relation to the objective of the paper) or are unclear:

“.. the phases of the self-sustained oscillation fluctuate.”

We modified the text to differentiate well-known facts from the main findings of this paper: “As expected, the phases of the self-sustained oscillations fluctuate³⁵⁻³⁷. Remarkably, we find that the phase fluctuations of the two modes are strongly anti-correlated, so that the measured sum of the two phases remains constant within our detection limit.”

“..our setup allows simultaneous measurement of the oscillation of both modes..”. How? Why is this relevant if each is connected to a separate actuator? How is your setup different than others?

We have modified the text to emphasize that “A crucial element of our experiment is that it is done in the sideband resolved regime. We find that self-sustained oscillations are induced not only in the low frequency mode as measured previously²²⁻²⁷, but also in the high frequency cavity mode. Our setup allows simultaneous measurement of the amplitude and phase of both modes, which provides a means to reveal the discrete time-translation symmetry of the system.”

The detection scheme as described in Methods shows how such measurement was implemented. In-plane motion of the beam is detected by measuring the current induced by the electromotive force in the magnetic field. For detecting motion of the plate, two fixed electrodes are fabricated underneath it. Vibrations of the top plate are detected by measuring the capacitance between it and the other electrode.

As our modified text indicates, simultaneous measurement is essential because one of our main findings is that the phase fluctuations of the two modes are anti-correlated. This measurement will not be possible if the oscillations of the two modes cannot be simultaneously detected.

Setups from other groups utilize piezoelectric and/or optical methods to measure the vibrations of the two modes.

“Without the external noise, the intrinsic eigenfrequency fluctuations of high-frequency (cavity) mode are the dominating source of phase fluctuations”, phase fluctuations of the plate or the beam mode? (for the beam mode this is obvious).

We have modified the text at the beginning of the paper to emphasize that the intrinsic eigenfrequency fluctuations are the dominating source of phase fluctuations. “Without the external noise, the intrinsic mode eigenfrequency fluctuations (the frequency noise) are the dominating source of phase fluctuations.”

Whether the eigenfrequency fluctuations of the beam or the plate dominate is discussed in detail later, under the section “Discussion”. We added the text “In our experiment, the anomalous phase diffusion arises mainly due to the frequency noise of the beam mode. Frequency noise of the plate mode is much smaller as its relatively large area allows for more efficient averaging of external disturbance such as forces due to trapped charges in dielectric layers.”

The two-mode electromechanical system

-In Figure 1, how the beam and plate modes have been excited? The ac signal is applied on which port (pump, probe1 or probe2) and at what frequency? The general explanation presented in *Methods-Transduction scheme* does not provide sufficient information.

We modified the text in Methods, which now reads

“For the beam mode, vibrations can be excited by applying an ac probe voltage at frequency close to ω_2 to an electrode next to the beam (Probe2 in Fig. 1a).”

“An ac probe voltage (Probe 1 in Fig. 1a) at frequency close to ω_1 applied to one of these electrodes (the lower one in Fig. 1a) can be used to capacitively generate a periodic electrostatic force.”

We have added the definition of $\Delta\omega_2$ used in the x label of Fig. 2a.

-“The parametric coupling between the plate mode and the phonon cavity are mediated by the tension in the beam as it deforms” What is the meaning of “mediation” here? Is this the same property explained in Ref. 29 (page 388 first sentence of the second paragraph: ...the motion of the first mode induces tension that can modify the frequency of the phonon cavity,....” Or a different mechanism is claimed?

We have modified the text to explain the parametric coupling more clearly.

“The parametric coupling between the plate mode and the phonon cavity (the beam mode) originates from the tension generated in the beam as it deforms parallel to the substrate, which in turn modifies the spring constant for the motion of the plate mode out of the substrate.”

-Here and throughout the two modes are referred to as high and low frequency modes, beam and plate modes and cavity mode. Please after the Figure-1 (where explain all terms) pick one set of words and stick to it.

We have changed high and low frequency modes into beam and plate modes.

-Clearly specify to which electrode (input port) you apply the ac current (instead of saying “applied to the beam” which implies exciting the beam mode.

The modified text reads “To parametrically pump the system near the sidebands, ac current is applied to the beam in a magnetic field to generate a periodic Lorentz force.” The current is not applied to the capacitive electrodes. The beam mode is not directly excited because the frequency of the pump current is far from the eigenfrequency of the beam mode and the linewidth of the beam mode is much smaller than the frequency detuning of the pump current.

-Clarify this sentence “.. this force leads to a displacement of the beam, changing the tension and hence...” parametrically modulating the mode coupling????

We added the clarification: “ This force leads to a displacement of the beam, which results in the direct parametric modulation of the mode coupling and also in an indirect modulation through the nonlinear coupling between the modes (see Supplementary Information).”

-Describe the physical nature of what you call “dispersive coupling coefficient” in the coupled system under test.

We have added the following text in the supplementary information:

“The first term on the right hand side of Eq. (S4) represents dispersive coupling in which the energy depends on the product of the squares of the beam and plate displacements.”

-What is the meaning of “driving the beam additively”?

We clarified in the text that additive driving refers to the 2nd term in Eq. S6. We modified the text below Eq. S6: “Here, the second term comes from the linear polarizability of the beam, whereas the first term, as explained in the main text, comes from the nonlinear response with the energy in the field being bilinear in the coordinates of the modes.”

Results

-Page 8 near the end of first paragraph: “the peak shifts with pumping current due to nonlinear effects”. Which nonlinear effects? Is this referring to nonlinear coupling between the two modes? The observed effect seems to be a manifestation of the so called “optical spring effect” in optomechanical resonator/oscillators. If that is the case who is this observation is related to the same effect in those systems?

We modified the text to specify the nonlinearity:

“The peak shifts with pumping current due to Duffing nonlinearity and dispersive coupling.”
The pumping does not produce any force gradients and therefore the observed effects are not related to the optical spring effect.

-What is physical relevance of ω_c (that only depends on damping and the amplitude of the pump) that makes it a critical parameter for bistable behavior?

We have put the expression for ω_c into a separate line, Eq. (3), and added description of the significance of ω_c .

“The frequency ω_c therefore serves as an important parameter that determines the range of detuning for bistable behavior. Interestingly, it is independent of the nonlinearity parameters.”

-It seems that one of the new observation here (and according to abstract) is self-sustained oscillation of both modes when stokes pump power is beyond certain value. As such a clear discussion about this result and its relevance is needed. Why this was not observed before? What characteristic of the current system makes this possible.

The observation of self-sustained oscillations of both modes allows us to study their phase diffusion and the strong anti-correlation. As mentioned before, we added the following text:

“A crucial element of our experiment is that it is done in the sideband resolved regime. We find that self-sustained oscillations are induced not only in the low frequency mode as measured previously²²⁻²⁷, but also in the high frequency cavity mode. Our setup allows simultaneous measurement of the amplitude and phase of both modes, which provides a means to reveal the discrete time-translation symmetry of the system.”

We also added the following paragraph:

“The presence of frequency noise has recently become one of the crucial factors that affect the performance of mechanical resonators and a number of schemes were developed to isolate the contribution of frequency noise from thermal or detector noise³⁸⁻⁴¹. Our results provide a novel way to identify and study frequency fluctuations via the phase diffusion of self-sustained oscillations. Furthermore, our findings of the strong phase anti-correlation of the two modes allow us to implement a feedback scheme that holds promise in significantly improving the phase stability of resonators in frequency standards and resonant sensing.”

Previous experiments in the resolved side-band limit only measured the low frequency mechanical mode but not the cavity mode. To emphasize the broader context of paper, we wrote in the last paragraph, “While our experiment uses a micromechanical resonator coupled to a phonon cavity, the analysis can be extended to optical and microwave cavities in the resolved sideband limit.’

It is possible to observe the self-sustained oscillations of both modes in our experiment because motion of the plate and the beam are detected separately and simultaneously: we detect the motion of the plate through its capacitive coupling to a nearby electrode and the motion of the beam through the back emf generated in a magnetic field.

-Why is it important to stabilize one mode using the feedback from measuring the phase fluctuations in another mode.

We have added a description of the importance of phase stabilization.

“The ability to generate vibrations with low phase noise is of paramount importance in mechanical oscillators, enabling sensitive resonant detection and high stability precision clocks.”

“The results enable tunable noise-free downconversion of the driving frequency ω_F to a frequency close to the eigenfrequency of the mechanical mode.”

-As at room temperature thermal noise is the dominant noise mechanism a discussion about expected behavior at room temperature is missing.

We added the following text:

“As the noise intensity or temperature increases, the phase fluctuations become stronger and D is expected to increase.”

End of Page 11) Why it has been assumed that even in the presence of external noise, the phases of the two modes are still anti-correlated?

We have added description in the text to explain that the phases of the two modes are anti-correlated because of discrete time translation symmetry. We have added Eq. S12 to show that without noise the total phase $\phi_1 + \phi_2$ is uniquely determined by the driving field frequency and amplitude. In the presence of noise, there are indeed fluctuations in $\phi_1 + \phi_2$. However, our analysis shows that these fluctuations do not accumulate in time and the mean value of $\phi_1 + \phi_2$ is given by Eq. S12.

The main weakness of the paper is that, at least in its current form, it appears as a characterization report with some physical insight. Several observations (some new and some just verification of previous work) have been presented in a scattered manner making the reader wonder about the main contribution of the authors. As such independent of its content the paper also requires major revision in writing style to become more accessible and understandable. I recommend the publication of the manuscript only if in a new revision the authors can clearly express the logic behind each experiment, relevance of their results and more importantly novelty of their approach/study compared to past work.

We are grateful to Reviewer 3 for this constructive criticism. We have followed the advice and significantly modified the paper to emphasize the rich physics behind our observations and why the experiments that we did were essential for revealing this physics. We hope that Reviewer 3, as well as Reviewers 1 and 2, will find that the new version of the paper meets the expectations and requirements for publication.

Reviewers' Comments:

Reviewer #2 (Remarks to the Author)

The main concern that I had during first review was what are the novelty and significance of this phase correlation phenomenon as there were likely to be quite similar results in the previous experiments. To this concern, they claimed and described such a way that this phenomenon occurs in "self-sustained oscillations" regime, the observation of which "has not been explored and theoretically", and is different from the previous works where "correlations in the oscillation quadrature of the two modes were observed as the quadratures fluctuate". Additionally they emphasized the significance as only this effect makes it possible to "observe phase diffusion" mechanism, namely that is impossible in the previous works. Thanks to this anti-phase correlation effect in the self-oscillations, the phase diffusion can be continuously monitored, thereby leading to development of the novel feedback technique. They took efforts to explain these significance and availability of this feedback technique in the revised manuscript, for example "...improving the phase stability of resonators in frequency standards and resonant sensing", "...enabling sensitive resonant detection and high stability precision clocks" and "...tunable noise-free downconversion of the driving frequency...".

Moreover, they carefully answered to the other questions and made necessary revisions which make non-expert readers easy to read and their misunderstandings to be possibly avoided. Finally, it would be nice if they can fully explore the underlying physics of $(1/f)^{0.7}$ noise by using the above novel technique, which will more clarify the availability of their technique and strengthen their work. But, as they mentioned, it "is beyond the scope of this paper" and therefore, I have no reason to pursue it any more.

Consequently, the authors clearly answered to my points and carefully revised the manuscript based on them, which supports the novelty and significance of this study. Therefore I don't have strong reasons to be against the publication in Nature Communications.

Reviewer #3 (Remarks to the Author)

I thank the authors for addressing all my concerns and suggestions. They have carefully responded to my comments as well as comments made by other reviewers. I am satisfied with their response and I recommend publication of the manuscript in its current form.